# Role of Noradrenaline and Adrenoreceptors in Regulating Prostaglandin E2 Synthesis Cascade in Inflamed Endometrium of Pigs

**DOI:** 10.3390/ijms24065856

**Published:** 2023-03-20

**Authors:** Barbara Jana, Jarosław Całka, Michał Bulc, Krzysztof Witek

**Affiliations:** 1Division of Reproductive Biology, Institute of Animal Reproduction and Food Research of the Polish Academy of Sciences, Tuwima 10, 10-078 Olsztyn, Poland; 2Department of Clinical Physiology, Faculty of Veterinary Medicine, University of Warmia and Mazury, Oczapowskiego 13, 10-718 Olsztyn, Poland

**Keywords:** endometrium, inflammation, noradrenaline, adrenoreceptors, prostaglandin E2 production/secretion, pig

## Abstract

In the inflamed uterus, the production and secretion of prostaglandins (PGs) and noradrenergic innervation pattern are changed. Receptor-based control of prostaglandin E2 (PGE2) production and secretion by noradrenaline during uterine inflammation is unknown. The aim of this study was to determine the role of α1-, α2- and β-adrenoreceptors (ARs) in noradrenaline-influenced PG-endoperoxidase synthase-2 (PTGS-2) and microsomal PTGE synthase-1 (mPTGES-1) protein levels in the inflamed pig endometrium, and in the secretion of PGE2 from this tissue. *E. coli* suspension (*E. coli* group) or saline (CON group) was injected into the uterine horns. Eight days later, severe acute endometritis developed in the *E. coli* group. Endometrial explants were incubated with noradrenaline and/or α1-, α2- and β-AR antagonists. In the CON group, noradrenaline did not significantly change PTGS-2 and mPTGES-1 protein expression and increased PGE2 secretion compared to the control values (untreated tissue). In the *E. coli* group, both enzyme expression and PGE2 release were stimulated by noradrenaline, and these values were higher versus the CON group. The antagonists of α1- and α2-AR isoforms and β-AR subtypes do not significantly alter the noradrenaline effect on PTGS-2 and mPTGES-1 protein levels in the CON group, compared to noradrenaline action alone. In this group, α1A-, α2B- and β2-AR antagonists partly eliminated noradrenaline-stimulated PGE2 release. Compared to the noradrenaline effect alone, α1A-, α1B-, α2A-, α2B-, β1-, β2- and β3-AR antagonists together with noradrenaline reduced PTGS-2 protein expression in the *E. coli* group. Such effects were also exerted in this group by α1A-, α1D-, α2A-, β2- and β3-AR antagonists with noradrenaline on mPTGES-1 protein levels. In the *E. coli* group, the antagonists of all isoforms of α1-ARs and subtypes of β-ARs as well as α2A-ARs together with noradrenaline decreased PGE2 secretion versus noradrenaline action alone. Summarizing, in the inflamed pig endometrium, α1(A, B)-, α2(A, B)- and β(1, 2, 3)-ARs mediate the noradrenaline stimulatory effect on PTGE-2 protein expression, while noradrenaline via α1(A, D)-, α2A- and β(2, 3)-ARs increases mPTGES-1 protein expression and α1(A, B, D)-, α2A- and β(1, 2, 3)-ARs are involved in PGE2 release. Data suggest that noradrenaline may indirectly affect the processes regulated by PGE2 by influencing its production. Pharmacological modulation of particular AR isoforms/subtypes can be used to change PGE2 synthesis/secretion to alleviate inflammation and improve uterine function.

## 1. Introduction

Endometritis and metritis are common and serious diseases in animals which lead to disturbances in reproductive processes and economic losses [1,2]. Bacteria are primarily responsible for the development of uterine inflammations. Difficult parturition and the retention of fetal membranes are factors that contribute to their occurrence [3]. Dysfunction of the immune system and/or uterine contractility determine the origin, development and maintenance of the inflammatory process [4,5]. Severe inflammation is associated with disturbances in uterine contractility, accumulation of inflammatory exudate in the uterine cavity and disorders of reproductive processes [6,7].

Uterine inflammation causes considerable changes in the rate of synthesis and secretion of many inflammatory mediators, including prostaglandins (PGs). In cows, high contents of PGE2 next to PGF2α were revealed in the uterine fluid during pyometra [8] and of PGE2 in the uterine secretion during endometritis [9]. The secretion of both PGs by endometrial cells was elevated in cows with clinical endometritis [10]. The pig endometritis increased the PGE2, PGF2α and PGI2 amounts in the endometrium and myometrium [11,12], and the endometrial expression of PG-endoperoxidase synthase-2 (PTGS-2, also known as PGHS-2 or COX-2) [11,13,14] and microsomal PTGE synthase-1 (mPTGES-1, also known as mPGES-1; enzyme down streaming PTGS-2, terminating PGE2 synthesis) [15]. PGE2, as a very important inflammatory mediator, plays a role in many processes leading to the classic signs of an inflammatory state [16]. In the cow endometrium, PGE2 increased the production and release of other pro-inflammatory mediators and damage-associated molecular patterns [17,18]. The role of this PG in the control of inflamed pig uterus contractility was also presented [19].

Catecholamines, including noradrenaline and adrenaline, act through adrenoreceptors (ARs) belonging to the seven transmembrane G protein-coupled receptor superfamily. There are two classes of AR: α and β. The α-AR class is divided into two subtypes: α1 and α2. Among these receptor subtypes, the following isoforms are identified: α1 A, B, D and α2 A, B, C. The β-AR class consists of three subtypes: 1, 2 and 3 [20]. The most numerous population of nerve fibers supplying pig uterus are catecholaminergic fibers and mainly express noradrenaline [21]. The presence of all α1- and α2-AR isoforms and all β-AR subtypes in the healthy uterus of pigs and other animal species has also been reported [22,23,24,25,26,27,28,29]. Catecholamines are involved in the regulation of uterine PGs secretion, among others, and α- and β-ARs participate in noradrenaline- and adrenaline-stimulated release of PGE2 (also PGF2α) from human myometrium [30] and in noradrenaline-induced PGE2 release from rat uteruses [31].

In the scope of the inflamed uterus, it is known that endometritis in pigs caused marked alterations in the uterine noradrenergic innervation pattern [32,33] and AR expression in inflamed endometrium [29]. In light of the above findings, it was hypothesized that noradrenaline, acting by α1- and α2-AR isoforms and β-AR subtypes, changes the PGE2 synthesis and secretion from the inflamed uterus. The receptor-based regulation of PGE2 production and secretion by noradrenaline during uterine inflammation remains unclear. Understanding these issues will significantly improve the prophylaxis and treatment of uterine inflammation in animals.

Therefore, the objectives of this study was to determine the influence of noradrenaline alone or together with antagonists of α1-, α2- and β-ARs on 1) the PTGS-2 and mPTGES-1 protein levels in the inflamed pig endometrium and 2) the secretion of PGE2 from this tissue.

## 2. Results

### 2.1. The Effect of Noradrenaline and/or Antagonists of α1-, α2- and β-ARs on the PTGS-2 Protein Levels in the Endometrium

#### 2.1.1. Noradrenaline Alone or Antagonists of α1-, α2- and β-ARs Alone

In the endometrium of the control (CON) group (gilts with saline injections into the uterus), the PTGS-2 protein levels did not differ significantly between the control value (untreated tissue) and in response to noradrenaline and all AR antagonists that were tested (Table 1). In the *Escherichia coli* (*E. coli*) group (gilts with *E. coli* injections into the uterus), noradrenaline increased the enzyme level, and it was higher versus the control value (*p* < 0.01), after using AR antagonists for: α1A (*p* < 0.001), α1B (*p* < 0.05), α1D (*p* < 0.001), α1ABD, α2A (*p* < 0.01), α2B, α1C, α2ABC (*p* < 0.001), β1 (*p* < 0.01), β2 (*p* < 0.05) and β3 (*p* < 0.01). In this group, the PTGS-2 levels were elevated in the untreated tissue (*p* < 0.05), in response to noradrenaline (*p* < 0.001), α1ABD-, α2A-, α2B- and β1-AR antagonists (*p* < 0.05) compared to the CON group.

#### 2.1.2. Noradrenaline Alone or Antagonists of α1-, α2- and β-ARs with Noradrenaline

Noradrenaline: Compared to the control values, noradrenaline did not significantly change the endometrial PTGS-2 protein level in the CON group, while it increased (*p* < 0.001) in the *E. coli* group (Figure 1). In this group, the PTGS-2 expression in the untreated tissue (*p* < 0.05) and after using noradrenaline (*p* < 0.001) was higher than in the CON group.

α1-AR antagonists with noradrenaline: In the CON group, the use of α1-AR antagonists in combination with noradrenaline did not result in a significant change in PTGS-2 protein levels relative to the control value or the effect of noradrenaline (Figure 1).The values of this enzyme in the *E. coli* group were lower in response to α1A-, α1B- (*p* < 0.001) and α1ABD (*p* < 0.01)-AR antagonists with noradrenaline than following added noradrenaline and α1D-AR antagonist with noradrenaline. Under the influence of α1ABD-AR antagonist with noradrenaline, the PTGS-2 expression was higher (*p* < 0.05) versus the control value and lower (*p* < 0.01) versus noradrenaline and α1D-AR antagonist with noradrenaline actions. Compared to the CON group, in the *E. coli* group endometrium, the PTGS-2 expression was increased after treatment with α1D- (*p* < 0.001) and α1ABD (*p* < 0.01)-AR antagonists with noradrenaline.

α2-AR antagonists with noradrenaline: All applied α2-AR antagonists with noradrenaline did not significantly alter the PTGS-2 protein expression in the CON group versus the control value and noradrenaline action (Figure 1). After using α2A-, α2C- and α2ABC-AR antagonists with noradrenaline in the *E. coli* group, the PTGS-2 amounts were decreased (*p* < 0.001) in relation to the effects of noradrenaline and α2B-AR antagonist with noradrenaline. The enzyme expression evoked by α2B-AR antagonist with noradrenaline was higher (*p* < 0.001) than the control value. In the *E. coli* group, compared to the CON group, the PTGS-2 levels were increased after using α2B- (*p* < 0.001) and α2ABC (*p* < 0.05)-AR antagonists with noradrenaline.

β-AR antagonists with noradrenaline: In the CON group, all used β-AR antagonists with noradrenaline did not significantly change the PTGS-2 protein levels versus the control value and noradrenaline influence (Figure 1). The PTGS-2 expression in the *E. coli* group was reduced by β1- and β2-AR antagonists with noradrenaline compared to the effects of noradrenaline (*p* < 0.001) and β3-AR antagonist with noradrenaline (*p* < 0.01). This enzyme level was increased by β3-AR antagonist with noradrenaline versus the control value (*p* < 0.05) and decreased (*p* < 0.05) in relation to the noradrenaline effect. In the *E. coli* group, the PTGS-2 protein level was higher in response to β3-AR antagonist with noradrenaline (*p* < 0.01) than in the CON group.

### 2.2. The Effect of Noradrenaline and/or Antagonists of α1-, α2- and β-ARs on the mPTGES-1 Protein Levels in the Endometrium

#### 2.2.1. Noradrenaline Alone or Antagonists of α1-, α2- and β-ARs Alone

In the CON group, the endometrial mPGES-1 protein expression did not differ significantly between the control value and after noradrenaline, and all used AR antagonist applications (Table 1). Noradrenaline, in the *E. coli* group, increased (*p* < 0.05) the mPTGES-1 expression compared to the control value and following the administration of all AR antagonists, except for α2A- and β2-ARs. The enzyme levels in the *E. coli* group were significantly higher in response to noradrenaline (*p* < 0.001) and α1ABD- and α2A-AR antagonists (*p* < 0.05).

#### 2.2.2. Noradrenaline Alone or Antagonists of α1-, α2- and β-ARs with Noradrenaline

Noradrenaline: The neurotransmitter did not significantly change the mPTGES-1 protein expression in the CON group, while it elevated (*p* < 0.001) this enzyme level in the *E. coli* group, and it was higher (*p* < 0.001) than in the CON group (Figure 2).

α1-AR antagonists with noradrenaline: In the CON group, all applied α1-AR antagonists with noradrenaline had no significant effect on the mPTGES-1 protein levels versus the control value and noradrenaline action (Figure 2). In the *E. coli* group, α1A- (*p* < 0.001), α1D- and α1ABD (*p* < 0.01)-AR antagonists with noradrenaline reduced the enzyme expression compared to noradrenaline and α1B-AR antagonist with noradrenaline influences. α1B-AR antagonist with noradrenaline increased (*p* < 0.001) the mPTGES-1 level versus the control value. The amount of mPTGES-1 in the *E. coli* group increased (*p* < 0.001) in response to the α1B-AR antagonist with noradrenaline compared to the CON group

α2-AR antagonists with noradrenaline: In the CON group, all used α2-AR antagonists with noradrenaline did not induce significant changes in mPTGES-1 expression compared to the control value and noradrenaline effect (Figure 2). In the *E. coli* group, the mPTGES-1 protein levels were higher after using α2A- (*p* < 0.01) and α2ABC (*p* < 0.001)-AR antagonists with noradrenaline than in response to noradrenaline, α2B- and α2C-AR antagonists with noradrenaline. The mPTGES-1 expression was higher (*p* < 0.001) after using α2B- and α2C-AR antagonists with noradrenaline versus the control value. In this group, the mPTGES-1 amounts were increased (*p* < 0.001) by α2B- and α2C-AR antagonists with noradrenaline compared to the CON group.

β-AR antagonists with noradrenaline: In the CON group, all applied β-AR antagonists with noradrenaline had no significant action on the mPTGES-1 protein levels versus the control value and noradrenaline action (Figure 2). This enzyme’s levels in the *E. coli* group were lowered (*p* < 0.001) by β2- and β3 -AR antagonists with noradrenaline versus noradrenaline and β1-AR antagonist with noradrenaline effects. Compared to the control value, the mPTGES-1 level was increased (*p* < 0.001) by β1-AR antagonist with noradrenaline. In the *E. coli* group, this enzyme expression was elevated by β1- (*p* < 0.001) and β2 (*p* < 0.05)-AR antagonists with noradrenaline versus the CON group.

### 2.3. The Effect of Noradrenaline and/or Antagonists of α1-, α2- and β-ARs on the Secretion of PGE2 from the Endometrium

#### 2.3.1. Noradrenaline Alone or Antagonists of α1-, α2- and β-ARs Alone

The CON group endometrium secreted more PGE2 in response to noradrenaline versus the control value (*p* < 0.01), and after using AR antagonists for: α1A (*p* < 0.05), α1B, α1D, α1ABD (*p* < 0.01), α2B, α1C, α2ABC (*p* < 0.05), β1, β2 and β3 (*p* < 0.01) (Table 2). In the *E. coli* group, the PGE2 release by noradrenaline was higher than the control value (*p* < 0.001) and evoked by AR antagonists for: α1A, α1B (*p* < 0.001), α1D, α1ABD, α2A, α1C, α2ABC (*p* < 0.01), β1, β2 and β3 (*p* < 0.001). In this group, noradrenaline elevated the PGE2 release more (*p* < 0.01) than in the CON group.

#### 2.3.2. Noradrenaline Alone or Antagonists of α1-, α2- and β-ARs with Noradrenaline

Noradrenaline: The neurotransmitter increased the PGE2 release in the CON (*p* < 0.01) and *E. coli* (*p* < 0.001) groups versus the control values (Figure 3). In the last group, the PG secretion was higher (*p* < 0.01) than in the CON group.

α1-AR antagonist with noradrenaline: In the CON group, the PGE2 release by α1A- (*p* < 0.01) and α1ABD (*p* < 0.05)-AR antagonists with noradrenaline was lower versus the effects of NA, α1B- and α1D-AR antagonists with noradrenaline (Figure 3). Under the influence of α1B- and α1D-AR antagonists with noradrenaline, the PGE2 secretion was higher (*p* < 0.01) than the control value. In the *E. coli* group, in response to α1A-, α1B-, α1D- and 1ABD-AR antagonists with noradrenaline, the PGE2 secretion was lower (*p* < 0.01) versus noradrenaline effect, and higher (*p* < 0.05) than the control value. Compared to the CON group, the PGE2 release in the *E. coli* group was increased (*p* < 0.01) by α1A-AR antagonist with noradrenaline.

α2-AR antagonists with noradrenaline: In the CON group endometrium, the PGE2 secretion was reduced (*p* < 0.05) by α2B- and α2ABC-AR antagonists with noradrenaline versus noradrenaline influence (Figure 3). α2A-AR antagonist with noradrenaline increased (*p* < 0.05) the PGE2 secretion compared to the control value. In the *E. coli* group, the PG release by α2A- and α2ABC-AR antagonists with noradrenaline was lower (*p* < 0.01) than noradrenaline action. The PGE2 secretion by α2A- (*p* < 0.01), α2B-, α2C- (*p* < 0.001) and α2ABC (*p* < 0.01)-AR antagonists with noradrenaline was greater compared to the control value. Larger (*p* < 0.01) PGE2 amounts were released in the *E. coli* group by α2B- and α2C-AR antagonists with noradrenaline versus the CON group.

β-AR antagonists with noradrenaline: The CON group endometrium released more (*p* < 0.01) PGE2 in response to β2-AR antagonist with noradrenaline versus noradrenaline and β1-AR antagonist with noradrenaline effects (Figure 3). The PG secretion was elevated by β1- (*p* < 0.01) and β3 (*p* < 0.05)-AR antagonists with noradrenaline compared to the control value. The PGE2 release in the *E. coli* group after treatment with β1-, β2- (*p* < 0.001) and β3 (*p* < 0.01)-AR antagonists with noradrenaline was higher than noradrenaline action. The PG secretion by β3-AR antagonist with noradrenaline increased (*p* < 0.05) compared to the control value. In the *E. coli* group, the PGE2 release by β1-AR antagonist with noradrenaline was lower (*p* < 0.01) than in the CON group.

## 3. Discussion

### 3.1. Our Findings in the Light of Available Literature

The neuronal mechanisms, including sympathetic, regulating the PG synthesis and secretion in the inflamed uterus are unknown. Endometritis in pigs increased the number of noradrenergic uterine perikarya in the caudal mesenteric [32] and paracervical [33] ganglia and changed the α1D-, α2 (A, C)- and β (1, 2)-AR amounts in the inflamed endometrium [29]. In the current study, we examined the receptor mechanisms by which the noradrenergic pathway alters the PGE2 synthesis and secretion from the inflamed pig endometrium. The results of the macroscopic and histopathological estimation of the uteri used in this study showed the presence of severe acute endometritis in the *E. coli*-treated uteri [29].

In this research, for the first time, noradrenaline’s role in the synthesis and secretion of PGE2 from both the healthy and inflamed endometrium has been confirmed. We found that noradrenaline elevated the PGE2 secretion from in the CON group endometrium. This agrees with research showing that under physiological conditions, noradrenaline enhanced the PGE2 release from the human myometrium [30], rat uterus [31], sow oviduct [34,35,36] and bovine luteal cells [37]. Outside the reproductive system, a similar noradrenaline action was revealed, for example, in rabbit gastric epithelial [38] and splenic [39] cells and rat mesenteric artery [40]. Our study shows that the noradrenaline-evoked increase in endometrial PGE2 secretion was accompanied by insignificantly altered PTGS-2 and mPTGES-1 protein expression. Noradrenaline also did not affect the PTGS-2 protein expression in rat primary microglia [41]. It is possible that the elevated PGE2 release by noradrenaline in the healthy endometrium may result from the activity of constitutively expressed PTGS-1, cytosolic PTGES (cPTGES) and mPTGES-2 localized in the uterus [15,42,43]. These types of PTGESs can couple with both PTGS-1 and PTGS-2. However, cPTGES may preferentially act in concert with PTGS-1 to maintain the PGE2 synthesis required for cellular homeostasis, while mPTGES-2 may be important for the PGE2 production involved in both tissue homeostasis and disease. During tissue inflammation or damage, the mPTGES-2 has a modest preference for coordination with PTGS-2 and is not increased appreciably [44,45].

We report that endometritis led to the increase in PGE2 secretion by noradrenaline, which coincides with the augmented PTGS-2 and mPTGES-1 protein expression. These parameters in the inflamed tissues were higher than in the healthy endometrium. The control value (untreated tissue) of PTGS-2 expression in the *E. coli* group was also elevated versus the CON group supporting earlier data [13,15]. Noradrenaline increased the PTGS-2 and PTGES mRNA/protein expression and the PGE2 release from human ovarian cancer cells [46] and PTGS-2 mRNA/protein expression and PGE2 secretion from lipopolysaccharide (LPS)-induced rat primary microglia [41]. Similarly, noradrenaline stimulated PGTS-2 mRNA/protein expression (also PGD2 secretion) in testicular macrophages of Syrian hamsters and non-testicular human macrophages (THP1 cell line) in studies on male infertility [47].

Entirely new are also the present indications about the importance of ARs in the noradrenaline influence on the synthesis and secretion of PGE2 from the healthy and inflamed uterus. To determine which kinds of ARs mediate noradrenaline effects on the endometrial PTGS-2 and mPTGES-1 protein expression and PGE2 release, the antagonists of particular isoforms of α1- and α2-ARs and subtypes of β-ARs were used. It was found that the application of the antagonists alone did not significantly change the values of examined enzyme expression and PG secretion in both groups versus the control values. In relation to noradrenaline action alone, the protein expression of the enzymes (the *E. coli* group) and PG release (both groups) were lower than the control values and in response to AR antagonists alone, except for the mPTGES-1 expression in response to α2A- and β2-AR antagonists in the *E. coli* group and the PGE2 release by α2A-AR antagonist in the CON group. The lack of significant effect of AR antagonists alone on PG secretion was revealed earlier in the rabbit gastric epithelial [38] and bovine endometrial [48] cells.

Our data show that the subtypes/isoforms of α1- and α2-ARs and subtypes of β-ARs did not participate in the noradrenaline action on the PTGE-2 and mPTGES-1 protein expression in the CON group endometrium. The mediation of α- and β-ARs in noradrenaline-stimulated PGE2 secretion has previously been established in both the human [30] and rat uterus [31]. The results of the current study further reinforce this mechanism by demonstrating the efficacy of blocking three isoforms of α1- or α2-AR subtypes. The current results, above all, significantly expand the earlier data showing the involvement of α1A- and α2B-ARs, as well as of β2-ARs in the PGE2 release from the healthy endometrium. The rise in PGE2 secretion after adrenergic actions was linked to the stimulation of α- and β-ARs in porcine oviduct [36], α1-ARs in gastric epithelial cells [38], α2-ARs in rabbit splenic pulpa [39] and β2-ARs in guinea pig trachea [49].

Literature data describing the receptor mechanisms of adrenergic stimulation in PGE2 synthesis/secretion during pathological states show that the activation of β2-ARs increased the PTGE-2 and PTGES expression and PGE2 release from human ovarian cancer cells [46]. The participation of β1- and β2-ARs in the enhancement of LPS-induced PTGE-2 protein expression in rat primary microglia [41] and this enzyme protein expression (also PGD2 secretion) in the testicular and non-testicular macrophages was also found [47]. In the *E. coli* group endometrium (the present study), we detected that the stimulation of β1-, β2- and β3-ARs by noradrenaline increased PTGE-2 protein expression and PGE2 secretion and β2- and β3-ARs enhanced the mPTGES-1 protein expression. These findings are generally in line with the above reports, indicating the additional role of β3-ARs. In the *E. coli* group, we observed activation of β1- and β3-ARs in addition to β2-ARs in response to noradrenaline, leading to an increase in PGE2 release. This is likely a result of elevated β1-AR mRNA and protein expression in the inflamed endometrium used in our study [29], as well as the differing distribution and ligand affinities of β1- and β3-ARs.

Earlier, the lack of involvement of α-ARs in the LPS-induced PTGE-2 protein expression in rat primary microglial cell culture [41] and in the testicular and non-testicular macrophages was revealed [47]. To the contrary, the total blocking of three isoforms of α1- or α2-AR subtypes shows that these subtypes mediate in the enhanced noradrenaline effects on PTGE-2 and mPTGES-1 protein expression and PGE2 release from the *E. coli* group endometrium. More precisely, we revealed that the noradrenaline stimulation of PTGS-2 protein expression involves α1A-, α1B-, α2A- and α2B-ARs, while α1A- α1D- and α2A-ARs are important for mPTGES-1 protein expression. All α1-AR isoforms and α2A-ARs are activated by noradrenaline to affect the PGE2 secretion. We also demonstrated that the partially different contribution of particular α-AR isoforms in noradrenaline influences this PG secretion in both studied groups (the CON group: α1A- and α2B-ARs, the *E. coli* group: all isoforms of α1-ARs and α2A-ARs). It should be added that in the inflamed endometrium, the mRNA/protein expression of α1D- and α2A-ARs was enhanced versus the healthy tissue [29]. As was suggested above, the localization of particular isoforms of α-ARs and their different ligand affinities may be relevant to the effects of adrenergic stimulation. The discrepancies in α-ARs’ involvement in the PTGS-2 expression in response to noradrenaline influence in the inflamed endometrium and earlier reports [41,47] can be related to the experimental model (cells, tissue explants) and kind of tissues. Considering the fact that α-ARs are largely located in vascular cells [29,50], the noradrenaline effect found on the PTGS-2 expression (also mPTGES-1 expression, PGE2 secretion) in the *E. coli* group endometrial explants could be due to changes in blood flow. It should also be added that downstream mechanisms of AR activation are not fully recognized. It was reported that noradrenaline could activate β2-ARs and transcriptionally activate PTGS-2 and PTGES via nuclear factor-kappaB (NF-kappaB) to produce PGE2 in human ovarian cancer cells [46]. The involvement of the mitogen-activated protein kinase (MAPK) signaling pathway in the effects of noradrenaline on LPS-induced PGE2 production and secretion in primary rat microglia has been proposed [41]. Further research is necessary to fully understand these relationships.

### 3.2. Clinical Significance of Findings

It is known that stressful factors activate the sympathetic nervous system which is linked to impaired female reproductive processes. Stress-related augmentation of circulating catecholamines may be able to promote inflammatory events. In the uterus, it may be related to PGE2 whose synthesis and secretion from the inflamed endometrium are elevated by noradrenaline (the present study). As mentioned earlier, the production/release of PGE2 increased in bovine [8,9,10,51] and porcine [11] uteri. During endometritis in pigs, this PG level in the utero-ovarian blood was decreased [11]. It has been established that PGE2 reduces the contractile amplitude and frequency of both kinds of uterine strips (endometrium with myometrium, myometrium alone) of the *E. coli*-infected uteri [19]. It is known that in bovine endometrial explants affected by LPS or *E. coli*, PGE2 production was increased via PTGS-2 and mPTGES-1, which led to the expression and secretion of pro-inflammatory mediators and damage-associated molecular patterns [17,18]. Pro-inflammatory cytokines (tumor necrosis factor α, interferon-γ) and leukotrienes (B4, C4) also led to the elevation in PGE2 release from the *E. coli*-treated cow uterine explants [52]. Our study not only expands knowledge on the regulation of PGE2 synthesis and secretion from the inflamed uterus but also describes the significant role of α2A-ARs, as well as three isoforms of α1- and three subtypes of β-ARs in mediating the effects of noradrenaline on PGE2 secretory activity in inflamed endometrium of pigs. These findings provide a basis for the development of drugs, such as antagonists targeting specific AR isoforms/subtypes. Blocking of AR isoforms/subtypes important for noradrenaline action on production/release of PGE2 could improve the secretory and contractile activities of inflammatory-changed uterus. For example, reducing the diastolic effect of PGE2 in an inflamed uterus during postpartum period (including porcine mastitis-metritis-agalactia syndrome) could improve the cleansing of the uterine cavity, supporting involution of the uterus.

It should be added that large amounts of PGE2 are associated also with LPS-induced pathological initiation of preterm labor [53,54]. Moreover, defects in PGE2 biosynthesis pathway are significant for the pathogenesis and symptoms of endometriosis [43]. Therefore, a decrease in PGE2 output by antagonists of ARs significant for noradrenaline influence in females with intrauterine infections during pregnancy could prevent preterm parturition and with endometriosis could have had a positive effect on the course of this pathology.

On the other hand, it is known that during inflammatory conditions, PGE2 acting locally together with PGI2 affects vascular permeability and pain generation [16]. Thus, application of AR isoforms/subtypes antagonists during uterine diseases associated with inflammation could modify the effect of PGE2 on the course of inflammatory process and, as a result, lead to its resolution.

Summarizing, antagonists of AR isoforms/subtypes indicated in the present study could be used in practice, in addition to currently administered drugs. As a result, it may contribute to more favorable outcomes in the prevention and treatment of uterine diseases associated with inflammation, particularly during the postpartum period, in animals as well as humans.

## 4. Materials and Methods

### 4.1. Animals

The Local Ethics Committee for Experiments on Animals (University of Warmia and Mazury in Olsztyn, Poland) approved all study procedures (Consent no. 65/2015). The guidelines in EU Directive 2010/63/EU for animal experiments were taken into account. The experiment was carried out on gilts (n = 10, Large White × Landrace, age 7–8 months, body weight 107.3 ± 1.8 kg/mean ± sem). A tester boar was used to detect behavioral estrus. The study gilts did not exhibit reproductive disorders (discharges from the reproductive tract were not observed, and the second estrous cycle took place regularly). The animals were transported from a commercial farm (Agro-Wronie Sp. z o.o., Wronie, Wąbrzeźno, Poland) to the local animal house (University of Warmia and Mazury, Olsztyn, Poland) three days prior to testing (acclimatization period). The animals stayed in individual pens (an area of about 5 m^2^) at a temperature of 18 ± 2 °C and the following light conditions: natural daylight—14.5 ± 1.5 h, night—9.5 ± 1.5 h. They received a commercial diet and had access to water.

### 4.2. Study Design

After the acclimatization period, the gilts were divided randomly (day 3 of the second estrous cycle—day 0 of the experiment) into two groups (each group consisted of 5 gilts): the CON (gilts with saline injections into the uterus) and the *E. coli* (gilts with *E. coli* injections into the uterus).

All study procedures were reported earlier [34]. Most importantly, after premedication (by atropine/Atropinum sulf. WZF, Warszawskie Zakłady Farmaceutyczne Polfa S.A., Poland, azaperone/Stresnil, Janssen Pharmaceutica, Beerse, Belgium, and ketamine hydrochloride/Ketamina, Biowet, Puławy, Poland) and general anesthesia (by ketamine hydrochloride), a median laparotomy was performed. In the *E. coli* group, 50 mL of *E. coli* suspension (content: 10^9^ cfu/mL, bacterial strain O25:K23/a/:H1; Department of Microbiology, National Veterinary Research Institute, Puławy, Poland) were injected into each uterine horn. In the CON group, 50 mL of saline solution was injected. Bacterial suspension and saline solution were injected into each horn in 5 places (10 mL/each place) at a similar distance from each other. All animals were untreated from the time of surgery to euthanasia. After eight days (the expected day 11 of the estrous cycle), the gilts were euthanized (by an overdose of sodium pentobarbital), and the uteri were collected.

### 4.3. Treatment of Endometrial Explants

To perform in vitro experiment, the uteri were transported on ice to the laboratory (within 20 min) and then washed twice with sterile phosphate-buffered saline (PBS, 137 mM NaCl, cat. no. 79412116, POCH, Poland, 27 mM KCl, cat. no. 739740114, POCH, Poland, 10 mM Na2HPO4, cat. no. 117992300, CHEMPUR, Piekary Śląskie, Poland, 2 mM KH2PO4, cat. no. 742020112, POCH, Poland; pH 7.4). A fragment of the uterine wall was collected from the middle part of each horn, and the endometrium was then separated from the myometrium by careful scraping using a scalpel blade. Afterwards, the endometrial fragments were cut and sliced (60–70 mg) and washed with Medium 199 (cat. no. M2520, Sigma, Saint Louis, MO, USA). Single explants of the endometrium were placed into glass vials containing 2 mL of Medium 199 supplemented with 0.1% BSA (cat. no. A2058, Sigma, Saint Louis, MO, USA) and antibiotics (500 μL/500 mL gentamicin, 100 μL/500 mL neomycin, cat. no. G1272, N1142, respectively, both from Sigma, Saint Louis, USA). The tissue fragments were preincubated and incubated in a shaking water bath at 37 °C in a humidified atmosphere of 95% air and 5% CO_2_. After a 1.5 h preincubation period, the endometrial explants were treated, for 16 h, with fresh (control) medium or with the addition of noradrenaline alone (10^−^^5^ M, Levonor, Warszawskie Zakłady Farmaceutyczne Polfa, Warszawa, Poland) or AR antagonists alone (each at a dose of 10^−^^4^ M) for: α1A- (RS 17053 hydrochloride, cat. no. 0985), α1B- (Rec 15/2615 dihydrochloride, cat. no. 3284), α1D- (BMY 7378 dihydrochloride, cat. no. 1006) and α1ABD (doxazosin mesylate, cat. no. 2964)-ARs, α2A- (BRL 44408 maleate, cat. no. 1133), α2B- (ARC 239 dihydrochloride, cat. no. 0928), α2C- (spiroxatrine, cat. no. 0631) and α2ABC (yohimbine hydrochloride, cat. no. 1127)-ARs, β1- (RS-atenolol, cat. no. 0387), β2- (ICI 118,551 hydrochloride, cat. no. 0821) and β3 (SR 59230A hydrochloride, cat. no. 1511)-ARs. The endometrial explants were also treated with noradrenaline (10^−^^5^ M) together with particular antagonists (each at a dose of 10^−^^4^ M). All antagonists were obtained from Tocris Bioscience. Initial dilutions of the used antagonists were done according to the manufacturer’s instructions (antagonists for: α1D-, α2A- α2B-, α2C-, β1- and β2-ARs were diluted in 0.2 mm filtrated distilled water; α1A-, α1B-, α1ABD-, α2ABC- and β3-ARs in dimethyl sulfoxide, cat. no. W387509, Sigma), and then stored at −20 °C. The final solutions of these agents, as well as the solution of noradrenaline, were prepared using the same medium for tissue preincubation and incubation. All treatments were performed in triplicate for each of the five gilts in the individual groups. A nitric oxide (NO) donor (NONOate; cat. no. 82150, Cayman Chemical Co., Ann Arbor, MI, USA) was used at 10^−^^4^ M as a positive control. The doses of noradrenaline, antagonists and NONOate, as well as the time of incubation, were determined in the preliminary studies or according to our previous research. After incubation, the endometrial fragments were blotted with a paper filter and then weighed and stored at −80 °C until analyzed for the expression of PTGS-2 and mPTGES-1 proteins. The medium was placed into tubes with 5% EDTA (cat. no. 118798103, Chempur, Piekary Śląskie, Poland), 1% acetylsalicylic acid (cat. no. 107140422, POCH Gliwice, Poland) solution (pH 7.4), and frozen at −20 °C for further determination of PGE2 concentration.

### 4.4. Western Blot Method

To estimate the expression of PTGS-2 and mPTGES-1 proteins in the endometrial explants, equal amounts of tissue fractions (20 μg) were dissolved in sodium dodecyl sulphate (SDS, cat. no. L3771, Sigma), a gel-loading buffer, heated (95 °C, 4 min) and separated by (10% for PTGS-2, 15% for mPTGES-1) SDS-polyacrylamide gel electrophoresis. Separated proteins were then electroblotted onto 0.2 µm pore size nitrocellulose membranes (cat. no. N2639, Whatman, Maidstone, UK) in a transfer buffer. The nonspecific binding sites were blocked by incubation with 5% fat-free dry milk (Spółdzielnia Mleczarska, Gostyń, Poland) in Tris (cat. no. T1503, Sigma)-buffered saline Tween 20 (cat. no. P1379, Sigma) buffer (21 °C, 1.5 h). After blocking, the nitrocellulose membranes were incubated (18 h, 4 °C) with polyclonal rabbit antibodies for PTGS-2 (in dilution 1:200; cat. no. 160107) and mPTGES-1 (in dilution 1:200; cat. no. 160140), both from Cayman Chemical Co. Subsequently, the nitrocellulose membranes for PTGS-2 study expression were incubated (21°C, 1.5 h) with biotinylated goat anti-rabbit IgG (in dilution 1:3000; Vectastain Elite ABC-HRP Kit, cat. no. PK-6101, Vector Labs, Newark, NJ, USA), incubated for 1 h, at 21 °C. Visualization of antibody binding was performed by incubation with a freshly prepared mixture of 3, 3`-diaminobenzidine tetrahydrochloride (DAB, cat. no. D5637, Sigma) and H2O2 in Tris-buffered saline (pH 7.2), for 3–4 min. In relation to mPTGES-1 expression, the nitrocellulose membranes were incubated (21 °C, 1.5 h) with secondary alkaline phosphatase-conjugated goat anti-rabbit antibody (in dilution 1:10,000; cat. no. NB7349, Novus Biologicals, Centennial, CO, USA). Immune complexes were visualized using a standard alkaline phosphatase visualization procedure (NBT-BCIP; cat. no. 72091, Sigma). Each analysis was repeated three times. To demonstrate the specificity of rabbit anti-PTGS-2 and anti-mPTGES-1 antibodies, specific binding peptides were used (cat. no. 360107, cat. no. 360140, respectively, both from Cayman Chemical Co.) as a negative control. After the neutralization of the primary antibodies for PTGS-2 and mPTGES-1, the bands were not found. The data for PTGS-2 were normalized in relation to the expression of glyceraldehyde-3-phosphate dehydrogenase (GAPDH) using primary polyclonal rabbit anti-GAPDH antibody (in dilution 1:5000; cat. no. G9545, Sigma), while data for mPTGES-1 were normalized in relation to the expression of β-actin (ACTB) using primary polyclonal rabbit anti-ACTB antibody (in dilution 1:6000; cat. no. ab119716, ABCAM). Images were gained and quantified by the use of a Quantity-One system (VersaDoc 4000M imaging system, Bio-Rad Laboratories, Hercules, CA, USA).

### 4.5. ELISA Procedure

The PGE2 concentration in the incubation medium was measured using the ELISA kits (cat. no. 514010, Cayman Chemical Co.), following the manufacturer’s protocols. The standard curve for PGE2 ranged from 7.8 to 1000 pg/mL. The intra- and interassay coefficients of variation were 4.7% and 6.2%, respectively.

### 4.6. Statistical Analysis

Only the results from the incubation of tissues for which the release of PGF2α in response to NONOate was statistically significant were considered. Normal distribution of all data (*p* > 0.05) was confirmed by Shapiro–Wilk test. A two-way ANOVA followed by the Bonferroni test (InStat Graph Pad, San Diego, CA, USA) was applied to compare the mean (±sem) values. The data are shown as mean (±sem) with differences accepted as statistically significant when *p* < 0.05.

## 5. Conclusions

This report shows that in the inflamed pig endometrium, α1(A, B)-, α2(A, B)- and β(1, 2, 3)-ARs mediate in the noradrenaline stimulatory influence on the PTGS-2 protein expression, while α1(A, D)-, α2A- and β(2, 3)-ARs are involved in this transmitter action on the mPTGES-1 protein expression. In turn, α1(A, B, D)-, α2A- and β(1, 2, 3)-ARs are activated by noradrenaline for the increase in PGE2 secretion. The above-mentioned ARs may play a crucial role in the effects of catecholamines on PGE2 production and secretion during spontaneous endometritis and metritis, affecting the severity of inflammatory symptoms and uterine function. Our results provide deeper insight into the pathomechanisms of bacterial endometritis, paving the way for the development of novel methods for the prevention and treatment of inflammatory diseases of uterus in both animals and humans.

## Figures and Tables

**Figure 1 ijms-24-05856-f001:**
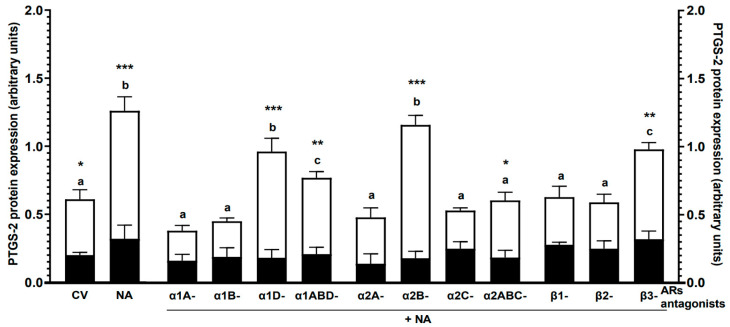
Effects of noradrenaline (NA, 10^−5^) alone or antagonists of α1-, α2- and β-ARs (10^−4^ M) with noradrenaline on the PTGS-2 protein expression in the pig endometrium of the CON (black bars) and *E. coli* (white bars) groups, estimated by the western blot method. Treatments were performed in triplicate for each of the five gilts in the individual groups. Values for PTGS-2 were normalized to GAPDH. The results are expressed as the mean ± sem. Blots with representative bands for each group are presented in Appendix A. Different letters (a–c) indicate statistical differences (*p* < 0.05–0.001) within each group for each subtype of α1- or α2-AR antagonists with noradrenaline or class of β-AR antagonists with noradrenaline and in relation to the control value (CV, untreated tissue) and noradrenaline action alone; *, **, *** indicate statistical differences (*p* < 0.05–0.001) between groups for the same treatment.

**Figure 2 ijms-24-05856-f002:**
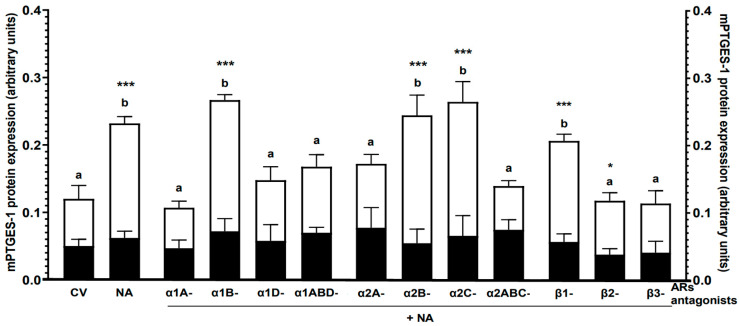
Effects of noradrenaline (NA, 10^−5^) alone or antagonists of α1-, α2- and β-ARs (10^−4^ M) with noradrenaline on the mPTGES-1 protein expression in the pig endometrium of the CON (black bars) and *E. coli* (white bars) groups, estimated by western blot method. Treatments were performed in triplicate for each of the five gilts in the individual groups. Values for mPTGES-1 were normalized to ACTB. Results are expressed as the mean ± sem. Blots with representative bands for each group are presented in Appendix A. Different letters (a, b) indicate statistical differences (*p* < 0.05–0.001) within each group for each subtype of α1- or α2-AR antagonists with noradrenaline or class of β-AR antagonists with noradrenaline and in relation to the control value (CV, untreated tissue) and noradrenaline action alone; *, *** indicate statistical differences (*p* < 0.05, *p* < 0.001) between groups for the same treatment.

**Figure 3 ijms-24-05856-f003:**
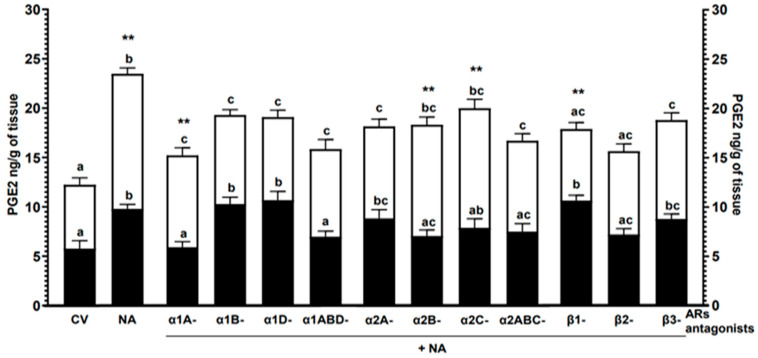
Effects of noradrenaline (NA, 10^−5^ M) alone or antagonists of α1-, α2- and β-ARs (10^−4^ M) with noradrenaline on the PGE2 secretion from the pig endometrium of the CON (black bars) and *E. coli* (white bars) groups, estimated by ELISA. Treatments were performed in triplicate for each of the five gilts in the individual groups. Results are expressed as the mean ± sem. Different letters (a–c) indicate statistical differences (*p* < 0.05–0.001) within each group for each subtype of α1- or α2-AR antagonists with noradrenaline or class of β-AR antagonists with noradrenaline in relation to the control value (CV, untreated tissue) and noradrenaline action alone; ** indicates statistical difference (*p* < 0.01) between groups for the same treatment.

**Table 1 ijms-24-05856-t001:** Effects of noradrenaline (10^−5^ M) alone or antagonists of α1-, α2- and β-ARs (10^−4^ M) alone on the PTGS-2 and mPTGES-1 protein expression in the pig endometrium of the CON and *E. coli* groups, estimated by western blot method. Treatments were performed in triplicate for each of the five gilts in the individual groups. Values for PTGS-2 were normalized to glyceraldehyde-3-phosphate dehydrogenase (GAPDH), and values for mPTGES-1 were normalized to β-actin (ACTB). Representative bands of PTGS-2 and mPTGES-1 protein expression in untreated tissues and in response to noradrenaline are presented in Appendix A, respectively. Data are expressed as the mean ± sem. Noradrenaline: NA; antagonist: anta.; control value: CV (untreated tissue).

Treatment	Group	Protein Expression (Arbitrary Units)	
		PTGS-2	mPTGES-1
CV	CON	0.2 ± 0.02	0.05 ± 0.01
	*E. coli*	0.41 ± 0.07 ^a,^*	0.07 ± 0.02 ^a^
NA	CON	0.32 ± 0.1	0.07 ± 0.01
	*E. coli*	0.94 ± 0.1 ^b,^***	0.17 ± 0.01 ^b,^***
α1A anta.	CON	0.18 ± 0.01	0.05 ± 0.01
	*E. coli*	0.23 ± 0.06 ^a^	0.09 ± 0.05 ^a^
α1B anta.	CON	0.32 ± 0.06	0.06 ± 0.01
	*E. coli*	0.48 ± 0.06 ^a^	0.09 ± 0.02 ^a^
α1D anta.	CON	0.22 ± 0.05	0.05 ± 0.01
	*E. coli*	0.24 ± 0.09 ^a^	0.08 ± 0.01^a^
α1ABD anta.	CON	0.21 ± 0.04	0.04 ± 0.01
	*E. coli*	0.37 ± 0.05 ^a,^*	0.09 ± 0.02 ^a,^*
α2A anta.	CON	0.2 ± 0.04	0.06 ± 0.02
	*E. coli*	0.43 ± 0.04 ^a,^*	0.11 ± 0.02 ^a,b,^*
α2B anta.	CON	0.2 ± 0.03	0.06 ± 0.01
	*E. coli*	0.34 ± 0.06 ^a,^*	0.07 ± 0.03 ^a^
α2C anta.	CON	0.21 ± 0.03	0.07 ± 0.01
	*E. coli*	0.29 ± 0.05 ^a^	0.07 ± 0.02 ^a^
α2ABC anta.	CON	0.26 ± 0.03	0.07 ± 0.02
	*E. coli*	0.38 ± 0.04 ^a^	0.07 ± 0.02 ^a^
β1 anta.	CON	0.23 ± 0.06	0.04 ± 0.01
	*E. coli*	0.47 ± 0.07 ^a,^*	0.04 ± 0.01 ^a^
β2 anta.	CON	0.31 ± 0.03	0.06 ± 0.02
	*E. coli*	0.52 ± 0.07 ^a^	0.1 ± 0.02 ^a,b^
β3 anta.	CON	0.24 ± 0.05	0.05 ± 0.01
	*E. coli*	0.41 ± 0.06 ^a^	0.06 ± 0.03 ^a^

Different letters (a, b) indicate statistical differences (*p* < 0.05–0.001) for each subtype of α1- or α2-AR antagonists or class of β-AR antagonists versus CV and noradrenaline effect for the same enzyme in the *E. coli* group (columns); *, *** indicate statistical differences (*p* < 0.05, *p* < 0.001) between groups for the same enzyme and treatment (rows).

**Table 2 ijms-24-05856-t002:** Effects of noradrenaline (10^−5^ M) alone or antagonists of α1-, α2- and β-ARs (10^−4^ M) alone on the PGE2 secretion from the pig endometrium of the CON and *E. coli* groups, estimated by ELISA. Treatments were performed in triplicate for each of the five gilts in the individual groups. Results are expressed as the mean ± sem. Noradrenaline: NA; antagonist: anta.; control value: CV (untreated tissue).

Treatment	Group	PGE2 (ng/g of Tissue)
CV	CON	5.77 ± 0.8 ^a^
	*E. coli*	6.46 ± 0.71 ^a^
NA	CON	9.8 ± 0.45 ^b^
	*E. coli*	13.67 ± 0.6 ^b,^**
α1A anta.	CON	6.95 ± 0.6 ^a^
	*E. coli*	7.15 ± 0.71 ^a^
α1B anta.	CON	6.23 ± 0.65 ^a^
	*E. coli*	6.08 ± 0.74 ^a^
α1D anta.	CON	6.69 ± 0.77 ^a^
	*E. coli*	8.01 ± 0.92 ^a^
α1ABD anta.	CON	6.58 ± 0.63 ^a^
	*E. coli*	8.68 ± 0.74 ^a^
α2A anta.	CON	7.61 ± 0.92 ^a,b^
	*E. coli*	6.28 ± 0.54 ^a^
α2B anta.	CON	6.58 ± 0.8 ^a^
	*E. coli*	8.6 ± 0.61 ^a^
α2C anta.	CON	7.18 ± 0.74 ^a^
	*E. coli*	6.54 ± 0.55 ^a^
α2ABC anta.	CON	7.41±0.89 ^a^
	*E. coli*	8.32 ± 0.92 ^a^
β1 anta.	CON	6.17 ± 0.38 ^a^
	*E. coli*	6.74 ± 0.77 ^a^
β2 anta.	CON	5.57 ± 0.63 ^a^
	*E. coli*	6.1±0.65 ^a^
β3 anta.	CON	6.08 ± 0.69 ^a^
	*E. coli*	6.74 ± 0.78 ^a^

Different letters (a, b) indicate statistical differences (*p* < 0.05–0.001) within each group for each subtype of α1- or α2-AR antagonists or class of β-AR antagonists versus CV and noradrenaline effect (column); ** indicates statistical difference (*p* < 0.01) between groups for the same treatment (rows).

## Data Availability

All relevant data are contained within the manuscript.

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
