# Peer review of "Role of Noradrenaline and Adrenoreceptors in Regulating Prostaglandin E2 Synthesis Cascade in Inflamed Endometrium of Pigs"

_ijms, 2023, doi:10.3390/ijms24065856_

Round 1

Reviewer 1 Report

Abstract.

The objectives of the study must be included in the abstract.

Introduction.

The final paragraph must be divided into two new ones: one to summarise the hypothesis of the authors and one to present clearly the objectives of the study.

Procedures.

Please include selection and inclusion criteria for animals.

How many days between transportation and start of the study?

For all products used in the study, please mention details of manufacturers.

Please provide evidence of normal distribution of data.

Results.

Figure 3. Please colourise.

Discussion.

Please divide in two sub-sections.

Also, please add a new sub-section about clinical significance of findings.

Author Response

The manuscript was corrected by native-speaker of English

Reviewer 1

Comments and Suggestions for Authors

Abstract.

The objectives of the study must be included in the abstract.

The objectives have been included (lines 15-18).

Introduction.

The final paragraph must be divided into two new ones: one to summarise the hypothesis of the authors and one to present clearly the objectives of the study.

It has been done (lines 76-87).

Procedures.

Please include selection and inclusion criteria for animals.

Information about animals is given in the Material and Methods part (lines 432-435).

How many days between transportation and start of the study?

This information is present in the manuscript (lines 435-438).

For all products used in the study, please mention details of manufacturers.

These data have been added (lines: 448-450,  456-459, 506, 510-512).

Please provide evidence of normal distribution of data.

Information about normal distribution of data is given (lines 543-544).

Results.

Figure 3. Please colourise.

We thank you for this suggestion. Nevertheless, please accept the present consistent colour scheme of Figures 1-3. It seems to us that it is the most readable. The introduction of colour involves additional printing costs.

Discussion.

Please divide in two sub-sections.

Also, please add a new sub-section about clinical significance of findings.

The Discussion part has been divided (lines 282 and 388).

Reviewer 2 Report

This is a straightforward study of the effects of noradrenalin, and adrenergic receptor agonists, on the expression of two prostaglandins in a pig model of uterine inflammation.  The difficulties and challenges associated with such experiments in pigs is appreciated.  However, a major concern arises from the use of the abundantly expressed actin protein as loading control in some of the Western Blots.  It would be good to re-probe using an antibody targeting a less abundant protein that does not produce such an overly saturated signal.  If tissue is available, it would be good to perform experimental replicates.

Overall, the paper seems not ready for publication, with figures missing or out of order, formatting issues, and generally needing a few more rounds of proofreading.  Perhaps the figure issues occurred during file processing, but the paper cannot be reviewed with figures lacking.  For such a straightforward story, the figures and tables should be easy to interpret, but they are not. 

In particular:

                  Line 12:  Spell out Prostaglandin E2 (PGE2) at first usage
                  Add commas in Line 57:  PGE2, as a very important inflammatory mediator, plays
                  Line 67:  add  “neurons” after catecholaminergic
                  Lines 69,70:  “Catecholamines are involved in regulation of uterine PGs secretion, among other activities.”
                  Line 76:  commas after “NA” and “subtypes”
                  Line 88:  Define “CON”  (1st use outside abstract)
                  Line 90:  Replace “all used ARs antagonists” with “all AR antagonists that were tested”
                  Line 91:  remove “and”

Section 2.2.2.  There are two treatments:  E. coli and noradrenaline.  Please be specific when mentioning treatment

Table I legend:  spell out gene names used for normalization.  Spell out other abbreviations used in table (CV)

Tables: 

arbitrary is misspelled. 
Rows are not all aligned.  Columns not aligned
Spacing surrounding +/- would improve readability. 
Listing the receptor as substitute for antagonist used is not the best.  Add the word agonist after the receptor type?

Even after consulting the Supplementary Material, it is not clear if these are technical replicates or experimental replicates.
From the tables, it looks as though antagonists were used alone, or together with E. coli.  Is this how the experiment was performed?

 The figures are not place properly--one seems to be missing and the two that are visible seem to be identical.  It would be best to plot the data into two bars rather than combining them.  This might make the statistics easier to interpret, because the statistics in the charts is confusing.

 Supplementary material

                  Please label sizes of MW standards
                  From the original Western Blots, the use of ACTB as a loading control was not a good choice as the protein is too abundant.  The Western signal for this protein is oversaturated.
                  The description of the samples in the figure legends is confusing.  It would be better to mark above each lane, as in a chart, the treatments for each sample.

Have the compounds been proven to act as antagonists in pigs?  References?

 Were all cells in the uterus that were used in the analysis equally exposed to the treatments?  How to be sure of this?

 Is heating in sample buffer really sufficient to disrupt the tissue?

 Suggested improvements:
                  Spell out noradrenaline each time

Author Response

The manuscript was corrected by native-speaker of English

Reviewer 2

Comments and Suggestions for Authors

This is a straightforward study of the effects of noradrenalin, and adrenergic receptor agonists, on the expression of two prostaglandins in a pig model of uterine inflammation.  The difficulties and challenges associated with such experiments in pigs is appreciated.  However, a major concern arises from the use of the abundantly expressed actin protein as loading control in some of the Western Blots.  It would be good to re-probe using an antibody targeting a less abundant protein that does not produce such an overly saturated signal.  If tissue is available, it would be good to perform experimental replicates.

Overall, the paper seems not ready for publication, with figures missing or out of order, formatting issues, and generally needing a few more rounds of proofreading.  Perhaps the figure issues occurred during file processing, but the paper cannot be reviewed with figures lacking.  For such a straightforward story, the figures and tables should be easy to interpret, but they are not.

In particular:

                  Line 12:  Spell out Prostaglandin E2 (PGE2) at first usage

It has been given (line 14).

                  Add commas in Line 57:  PGE2, as a very important inflammatory mediator, plays

It has been changed (line 59).

                  Line 67:  add  “neurons” after catecholaminergic

This sentence has been rewritten. We added word ‘fibers’ (line 69).

                  Lines 69,70:  “Catecholamines are involved in regulation of uterine PGs secretion, among other activities.”

This sentence has been changed (line 72).

                  Line 76:  commas after “NA” and “subtypes”

It has been improved (line 79).

                  Line 88:  Define “CON”  (1st use outside abstract)

The “CON” has been defined (line 92). We explained also E. coli group (line 95).

                  Line 90:  Replace “all used ARs antagonists” with “all AR antagonists that were tested”

This sentence has been changed (lines 94 and 95).

                  Line 91:  remove “and”

It has been improved (line 97).

Section 2.2.2.  There are two treatments:  E. coli and noradrenaline.  Please be specific when mentioning treatment

In this section we give data concerning two treatments:

  • noradrenaline action alone, and
  • AR antagonists together with noradrenaline

The subsection names have been introduced to make it easier to track the results (section 2.2.2). The changes have been also made in sections: 2.1.2 and 2.3.2.

Table I legend:  spell out gene names used for normalization.  Spell out other abbreviations used in table (CV)

These changes have been included.

Tables:

arbitrary is misspelled.

It has been improved.

Rows are not all aligned.  Columns not aligned

The Tables 1 and 2 have been improved.

Spacing surrounding +/- would improve readability.

It has been given.

Listing the receptor as substitute for antagonist used is not the best.  Add the word agonist after the receptor type?

Into the Tables we added the abbreviation for antagonist “anta.”

Even after consulting the Supplementary Material, it is not clear if these are technical replicates or experimental replicates.

Into the description of Tables and Figures we added following sentence “Treatments were performed in triplicates for each of 5 gilts in the individual groups”.

From the tables, it looks as though antagonists were used alone, or together with E. coli.  Is this how the experiment was performed?

For the avoidance of doubts, in the results section, when the names of the groups (CON, E. coli) first appear, we give a brief experimental procedure.

Uterine sections from these groups were treated:

- the antagonists alone

- noradrenaline alone

- antagonists together with noradrenaline.

The Tables show the effect of the antagonists alone and noradrenaline alone on the parameters studied.

The Figures show the effects of antagonists together with noradrenaline, as well as the effect of noradrenaline alone on the parameters studied.

The description of the experiment is extensively given in the Materials and Methods part.

The figures are not place properly--one seems to be missing and the two that are visible seem to be identical. It would be best to plot the data into two bars rather than combining them.  This might make the statistics easier to interpret, because the statistics in the charts is confusing.

We removed the Figure that was double.

We tried different types of graphical charts to represent the results. Please believe us that due to the large amount of data, this form of graph is the clearest to represent the results.

Supplementary material

                  Please label sizes of MW standards

MW standards have been added.

                  From the original Western Blots, the use of ACTB as a loading control was not a good choice as the protein is too abundant.  The Western signal for this protein is oversaturated.

We agree with the Reviewer that ACTB protein signal is oversaturated, and antibody for protein with less abundance should be used. Unfortunately, we cannot repeat the experiment due to the lack of tissue. The in vitro study was performed on endometrial sections. We thank the Reviewer for pointing out the protein expression of ACTB. We will take this remark into account in future studies.

                  The description of the samples in the figure legends is confusing.  It would be better to mark above each lane, as in a chart, the treatments for each sample.

The description of the samples on the blots have been changed similarly to the description in Figures.

Have the compounds been proven to act as antagonists in pigs?  References?

The same antagonists as in the current study, were used by us in pigs to determine the receptor mechanism of the noradrenaline influence on the contractility of inflamed and healthy uteri. These antagonists significantly changed the contractile parameters in uterine sections (myometrium, endometrium with myometrium) induced by noradrenaline.

References:

  1. Jana B., Całka J., Role of beta-adrenergic receptor subtypes in pig uterus contractility with inflammation. Sci. Rep. 11(1) (2021) 11512.
  2. Jana B., Całka J., Bulc M., Roles of alpha-2-adrenergic receptor isoforms in inflamed pig uterus contractility in vitro. Theriogenology 183 (2022) 41-52.
  3. Jana B., Całka J., Effect of blocking of alpha1-adrenoreceptor isoforms on the noradrenaline-induced changes in contractility of inflamed pig uterus. PLOS ONE, 2023, in press

Were all cells in the uterus that were used in the analysis equally exposed to the treatments?  How to be sure of this?

The study was performed on the endometrial explains composed of epithelial (luminal, glandular), stromal and blood vessel (endothelium, muscular layer) cells. The immunofluorescent study of the gilt uteri showed a distribution of all ARs in these cells (Meller K.A., Całka J., Kaczmarek M., Jana B. Expression of alpha and beta adrenergic receptors in the pig uterus during inflammation. Theriogenology 2018, 119, 96–104). Moreover, in the used endometrial explants were present immunocompetent cells, particularly in the inflamed endometrium. It was found that immune cells, including neutrophils, may be fundamental locations of ARs receptor expression (LaBranche T.P., Ehrich M.F., Eyre P. Characterization of bovine neutrophil beta2-adrenergic receptor function. J Vet Pharmacol Ther 2010,33, 323-31; Brunskole Hummel I., Reinartz M.T., Kälble S., Burhenne H., Schwede F., Buschauer A., Seifert R. Dissociations in the effects of β2-adrenergic receptor agonists on cAMP formation and superoxide production in human neutrophils: support for the concept of functional selectivity. PLoS One 2013, 31;8(5):e64556. doi: 10.1371/journal.pone.0064556).

Thus, to answer the question in which degrees the particular kinds of endometrial cells response to treatments, the in vitro studies on isolated cells should be performed.

Is heating in sample buffer really sufficient to disrupt the tissue?

The procedure is sufficient and widely used in laboratories. 

Suggested improvements:

                  Spell out noradrenaline each time

It has been given in the whole manuscript.

Round 2

Reviewer 1 Report

The authors have improved the manuscript.

Before acceptance, I recommend the addition of a brief passage in the discussion about the clinical importance of this study.

Author Response

Reviewer 1

The authors have improved the manuscript.

Before acceptance, I recommend the addition of a brief passage in the discussion about the clinical importance of this study.

In the Discussion there is the section on the supposed practical significance of the research. As  Reviewer suggested, we have emphasized this even more (lines 422-429, line 560).

Reviewer 2 Report

In the text of the paper, there are no referrals to figures.  This should be corrected so that the reader can view the data while following along with the text.

Table I should refer to the appropriate figure in Supplementary Information. 

With regard to Table I, when reporting statistical significance differences, the two points in question are typically highlighted by connecting bars, and this is also the case for the figures.  Please highlight the values being compared for statistical significance.

With regard to methodology, how the explants are incubated in the antagonist experiments should be better described.  Is noradrenalin also being included in the incubations involving antagonists (CON and E. coli samples)? The “and/or” terminology used does not help clarify and the table is not clearly labeled.

This is a straightforward study but there are concerns regarding how the data is presented.  There is no description of Figures 1, 2 or 3 in the text, and it is concerning that the values in each of these plots does not match the values in their corresponding data tables.  If any calculations were performed before plotting, this should be explained.  The figures could be plotted in such a way to better represent the data.  The axes values are difficult to line up with the data values. Stacked columns may not the best way to represent the data, but perhaps this is just personal preference.

Author Response

Reviewer 2

In the text of the paper, there are no referrals to figures. This should be corrected so that the reader can view the data while following along with the text.

The referrals to figures were given by us in the text (to Fig. 1 - lines: 119, 123, 134, 143; to Fig. 2 – lines: 192, 195, 203, 212; to Fig. 3 – lines: 248, 257, 266,). Only in one place was there no reference to Fig. 3. This has been added (line 244)

Table I should refer to the appropriate figure in Supplementary Information.

We want to clarify that Table 1 shows the expression of enzymes:

  - in untreated tissue (CV),

- in response to noradrenaline alone,

- in response to the particular antagonists of adrenoreceptors alone.

In the supplementary materials (Figures 1 and 2) we do not present representative blots for the effects of particular antagonists of adrenoreceptors alone.

The supplementary materials (Figures 1 and 2) show blots for enzyme expression:

- in untreated tissue (CV),

- in response to noradrenaline alone,

- in response to particular antagonists of adrenoreceptors together with noradrenaline.

Thus, supplementary Figures 1 and 2 refer to Figures 1 and 2 in the text.

The common part of Table 1 and Supplementary Figures 1 and 2 relates only to enzyme expression in untreated tissues and in response to noradrenaline. To the headline of Table 1 we added that „Representative bands of PTGS-2 and mPTGES-1 protein expression in untreated tissues and in response to noradrenaline are presented in Supplementary Figures 1 and 2, respectively”.  

With regard to Table I, when reporting statistical significance differences, the two points in question are typically highlighted by connecting bars, and this is also the case for the figures.  Please highlight the values being compared for statistical significance.

To facilitate undertanding the statistical analysis into legends of  Tables 1 and 2 we added indications of „columns” and „rows”. The ststistically significant differences on Figures 1, 2 and 3 show:

  • different letters - within each group for each subtype of α1- or α2-ARs antagonists with noradrenaline or class of β-ARs antagonists with noradrenaline and in relation to the control value (CV, untreated tissue) and noradrenaline action alone;
  • x - between groups for the same treatment.

With regard to methodology, how the explants are incubated in the antagonist experiments should be better described.  Is noradrenalin also being included in the incubations involving antagonists (CON and E. coli samples)? The “and/or” terminology used does not help clarify and the table is not clearly labeled.

We explain that explants (from the Control and E. coli groups) were incubated with:

- particular antagonists of adrenoreceptors alone,

- noradrenaline alone,

- particular antagonists of adrenoreceptors together with noradrenaline.

This pattern of study justifies the use of the term „and/or”

To improve understanding of the methodology, we changed the description of treatment of endometrial explants (lines 478-489).

Moreover, into the headlines of Table 1 and Table 2 as well as into the subtittles (2.1.1, 2.2.1, 2.3.1) we added „alone” what will make the results easier to read.

This is a straightforward study but there are concerns regarding how the data is presented.  There is no description of Figures 1, 2 or 3 in the text, and it is concerning that the values in each of these plots does not match the values in their corresponding data tables.  If any calculations were performed before plotting, this should be explained.  The figures could be plotted in such a way to better represent the data.  The axes values are difficult to line up with the data values. Stacked columns may not the best way to represent the data, but perhaps this is just personal preference.

We explain that the values in Tables 1 and 2 concern influences of particular antagonists of adrenoreceptors alone, while the values on Figures 1, 2 and 3 concern actions of adrenoreceptors antagonists together with noradrenaline. The same values concern only untreated tissue (CV) and noradrenaline action alone.

We agree that our way of data presentation in Figures 1, 2 and 3 is slightly complicated but on the other hand presentation of separate columns for the Control and E. coli groups would generate additional 12 columns making it even more difficult for graphic presentation. Therefore we would appreciate acceptance of our way of the data presentation.
